

# Wing morphological responses to latitude and colonisation in a range expanding butterfly

Evelyn D. Taylor-Cox[1], Callum J. Macgregor[2,3], Amy Corthine[1], Jane K. Hill[2], Jenny A. Hodgson[1] and Ilik J. Saccheri[1]

[1] Department of Evolution, Ecology and Behaviour, University of Liverpool, Liverpool, United Kingdom
[2] Leverhulme Centre for Anthropocene Biodiversity, Department of Biology, University of York, York, United Kingdom
[3] Energy and Environment Institute, University of Hull, Hull, United Kingdom

Corresponding author
Evelyn D. Taylor-Cox,
e.taylorcox@hotmail.co.uk

## ABSTRACT

Populations undergoing rapid climate-driven range expansion experience distinct selection regimes dominated both by increased dispersal at the leading edges and steep environmental gradients. Characterisation of traits associated with such expansions provides insight into the selection pressures and evolutionary constraints that shape demographic and evolutionary responses. Here we investigate patterns in three components of wing morphology (size, shape, colour) often linked to dispersal ability and thermoregulation, along latitudinal gradients of range expansion in the Speckled Wood butterfly (*Pararge aegeria*) in Britain (two regions of expansion in England and Scotland). We measured 774 males from 54 sites spanning 799 km with a 10-year mean average temperature gradient of 4 °C. A geometric morphometric method was used to investigate variation in size and shape of forewings and hindwings; colour, pattern, and contrast of the wings were examined using a measure of lightness (inverse degree of melanism). Overall, wing size increased with latitude by ~2% per 100 km, consistent with Bergmann's rule. Forewings became more rounded and hindwings more elongated with history of colonisation, possibly reflecting selection for increased dispersal ability. Contrary to thermal melanism expectations, wing colour was lighter where larvae developed at cooler temperatures and unrelated to long-term temperature. Changes in wing spot pattern were also detected. High heterogeneity in variance among sites for all of the traits studied may reflect evolutionary time-lags and genetic drift due to colonisation of new habitats. Our study suggests that temperature-sensitive plastic responses for size and colour interact with selection for dispersal traits (wing size and shape). Whilst the plastic and evolutionary responses may in some cases act antagonistically, the rapid expansion of *P. aegeria* implies an overall reinforcing effect between these two mechanisms.

## INTRODUCTION

A population may respond to climate change either by altering its phenotype to maintain local fitness, or by shifting distribution and/or phenology to track its climatic envelope (*Parmesan & Yohe, 2003*; *Macgregor et al., 2019*). During a range expansion populations on the range front are subject to multiple selection pressures (*Phillips, Brown & Shine, 2010*). Any resulting phenotypic changes are therefore created by evolutionary responses to both changing local environments and the process of range expansion itself. Phenotype-environment optima under equilibrium conditions, may be overridden or obscured by the expansion process as mal-adapted genotypes can surf on the range front due to genetic drift (*Burton & Travis, 2008*). Moreover, temperature-dependent reaction norms, and genetic correlations among traits, may help or hinder adaptation to the new environment (*Pujol et al., 2018*). Phenotypic responses to climate change have been reported for correlates of dispersal (*Thomas et al., 2001*; *Hill, Griffiths & Thomas, 2011*), body size (*Daufresne, Lengfellner & Sommer, 2009*), and colour lightness (linked to thermal tolerance; *Zeuss et al., 2014*).

Contemporary evolution of dispersal traits is often closely linked to the process of range shifts towards cooler climates (*Parmesan, 2006*; *Hickling et al., 2006*). Indeed, a common phenotypic signature of range expansion, found in many species, is increased dispersal ability towards the leading edge of a shifting range (*Hughes, Hill & Dytham, 2003*; *Simmons & Thomas, 2004*; *Phillips, Anderson & Schapire, 2006*). This cline in dispersal ability is a product of *spatial selection*, resulting from the combined effects of spatial sorting (assortative mating of dispersive genotypes on the range edge) and density-dependent selection (*Phillips, Brown & Shine, 2010*). Density-dependent selection is predicted to favour good dispersers colonising new habitat patches, where they encounter much lower intraspecific competition, resulting in increased fecundity and intrinsic growth rates at the range edge. Dispersal in insects, of which flight performance is a key component, may be affected by several factors including morphological, physiological, metabolic, and behavioural traits (*Betts & Wootton, 1988*; *Berwaerts, Van Dyck & Aerts, 2002*; *Niitepõld et al., 2009*; *Flockhart et al., 2017*; *Renault, 2020*). Thorax size (highly correlated to whole body size) is a widely accepted measure of dispersal ability in Lepidoptera as it indicates flight muscle investment (*Srygley & Chai, 1990*; *Hill, Thomas & Lewis, 1999*; *Berwaerts, Van Dyck & Aerts, 2002*). Wing size has been shown to be correlated with body size measurements (e.g., thorax size, body length and dry mass) in several Lepidoptera species (*Chai & Srygley, 1990*; *Merckx & Van Dyck, 2006*) and hence can be used as a proxy for overall size and dispersal (*Sekar, 2012*). Wing shape has direct implications for aerodynamics during flight, which affects the efficiency of flight strategies, flight ability and dispersal (*Breuker, Brakefield & Gibbs, 2007*; *Le Roy, Debat & Llaurens, 2019*).

Morphological trends associated with poleward range expansion could be due to spatial selection for dispersal ability, but could also reflect a genetic or plastic response to an environmental cline. One such phenotype-environment relationship is an increase in body size with latitude (Bergmann's rule; *Bergmann, 1848*). Originally described in mammals, this pattern (and its converse) has been observed in ectotherms (*Shelomi, 2012*).

Bergmann clines often suggest genetic adaptation to different thermal environments but phenotypic plasticity also plays an important role in producing body size clines (*Atkinson & Sibly, 1997*; *Mousseau, 2006*). The temperature size rule, describes the plastic response of body size to developmental temperature (smaller individuals at higher temperatures) in ectotherms (*Atkinson, 1994*). However, selection for increased size at lower temperature is not supported across all species (*Stillwell, Moya-lara & Fox, 2008*), and the relationship of size to temperature may evolve in different directions between recently divergent populations that experience contrasting temperature regimes (*Kingsolver et al., 2007*). Body size clines in ectotherms are further affected by season length and voltinism, i.e., the number of generations per year, as growing time is positively correlated to size (*Chown & Gaston, 2010*; *Horne, Hirst & Atkinson, 2015*; *Zeuss, Brunzel & Brandl, 2017*). Above a certain latitude, shortening of season length reaches a point where the limited time available for development, growth and foraging results in reduced overall body size with latitude, i.e., a converse Bergmann cline (*Blanckenhorn & Demont, 2004*). The addition of a second-generation can result in complex saw-tooth patterns of body size with season length (*Roff, 1980*). Season length and voltinism may therefore explain why both the inverse and classical Bergmann's rule have been documented in arthropods (*Horne, Hirst & Atkinson, 2015*), suggesting that these patterns are not contradictory but part of a continuum (*Blanckenhorn & Demont, 2004*).

Melanism is another trait often associated with adaptation to the thermal environment (thermal melanism). For example, in ectotherms darker species frequently occur at higher latitudes or in cooler climates (*Zeuss et al., 2014*; *Heidrich et al., 2018*). This observation is often explained by the thermal melanism hypothesis, which states that darker individuals have an advantage in cooler climates (*Clusella Trullas, Van Wyk & Spotila, 2007*). In principle, all other things being equal, ectotherms presenting a larger and/or darker surface area of melanised exocuticle should show increased absorption of solar radiation compared to lighter individuals, thus reaching a higher body temperature and at a faster rate. This could, in turn, allow activity at lower temperatures, potentially enhancing mating opportunities (*Clusella Trullas, Van Wyk & Spotila, 2007*) and dispersal (*Mattila, 2015*). It has been suggested that the basal part of the wing is the most important for thermal regulation (*Wasserthal, 1975*), but other components of the wing pattern may contribute to the thermal properties of wings (*Brashears, Aiello & Seymoure, 2016*). Melanism also plays an important role in protection against UV radiation (*Bishop et al., 2016*; *Katoh, Tatsuta & Tsuji, 2018*) and pathogens (*Dubovskiy et al., 2013*), which may lead to darker individuals in warmer climates, opposing the trend predicted by a purely thermal explanation. The degree of melanism may also be affected by selection on the colour pattern which has important functions in mate choice (*Jiggins et al., 2001*; *Kemp, 2007*) and predation avoidance (*Bond & Kamil, 2002*). Futhermore, seasonal polyphenism (the production of different phenotypes in different seasonal generations) is widely documented in Lepidoptera, and particularly prominent in multivoltine species (*Kingsolver, 1995*). This phenomenon is driven by environmental cues (*Roskam & Brakefield, 1999*), often altering wing pattern, which can potentially produce pattern differences across environmental gradients.

In this study, the Speckled Wood butterfly (*Pararge aegeria* (Linnaeus, 1758)), which has undergone rapid range expansion in mainland Britain, was used to investigate phenotypic changes in wing size, shape and melanism, with respect to colonisation history, latitude and temperature. Flight morphological traits in *P. aegeria* differ across latitudinal clines in mainland Europe, by habitat type (*Van dewoestijne & vanDyck, 2010*) and with mate location strategy (*Berwaerts, Van Dyck & Aerts, 2002*). Bergmann's rule has been reported for the British population, with larger individuals found further north (*Dennis & Shreeve, 1989*; *Sibly, Winokur & Smith, 1997*), whereas the inverse pattern was seen in Sweden (*Nylin & Svärd, 1991*). An increase in dispersal ability, using thorax size as an indicator, was found towards the expanding range edge in the UK, with a potential evolutionary trade-off between reproduction and dispersal (*Hughes, Hill & Dytham, 2003*). These studies of the British populations were limited to a relatively small number of sites, when the distribution of *P. aegeria* in England and Scotland was much less extensive. Detailed analysis of *P. aegeria* wing morphology, especially using geometric morphometrics, across the range expansion in Britain is lacking. *Dennis & Shreeve (1989)* report latitudinal variation in *P. aegeria* wing colour which is consistent with the anecdotal perception of butterfly recorders that individuals from northerly populations tend to be darker than those from southern populations. The cream spot pattern of *P. aegeria* has been described to increase in lightness and size with latitude, possibly due to the interplay between thermoregulatory requirements, mate choice and predator escape (*Dennis & Shreeve, 1989*). However, to our knowledge, the qualitative perceptions of *P. aegeria* colour and pattern have not been verified quantitatively.

*Pararge aegeria* in Britain provides an opportunity to examine the interplay and influence of demographic factors (range expansion) and environmental factors (latitude and temperature, both during development and in the recent past) in shaping morphological traits in a rapidly expanding population. We investigated phenotypic changes in wing size and shape (linked to dispersal ability and body size), as well as colour and pattern (potentially influencing thermoregulation) across the expanded range of *P. aegeria* in mainland Britain. In common with similar studies on wild-caught individuals that lack experimental and/or genetic data, our sample does not allow strong inferences about the relative importance of phenotypic plasticity vs. genotypic differences in determining the observed patterns in morphological variation. The relative effect of demographic and environmental factors on morphology were evaluated through collection of samples along the axes of range expansion, specifically to capture a wide range of local population ages and colonisation histories (from core sites known to have been continuously occupied at least since 1965 to leading edge sites colonised in 2015), as well as over large gradients in latitude (covering a distance of 799 km), mean 10-year annual temperature (4 °C) and mean temperature during development (6 °C). We hypothesised that: (1) wing size is larger in more recently colonised populations, as predicted by spatial selection; (2) wing size increases with latitude, following Bergmann's and the temperature-size rules; (3) wing size is smaller in populations with more generations; (4) wing shape changes to a more dispersive form with colonisation history; (5) melanism increases with latitude and decreasing temperatures, in accordance with the thermal melanism hypothesis; and (6) the

cream spot wing pattern becomes lighter and larger with latitude, as described by *Dennis & Shreeve (1989)*.

## MATERIALS & METHODS

### Study species and sample collection

*Pararge aegeria* is a multivoltine species (capable of completing multiple generations in a year) which occurs throughout Europe to western Asia. *Pararge aegeria* can follow three different developmental pathways—pupal diapause, larval diapause or direct development—that may differentially affect morphology (*Van Dyck & Wiklund, 2002*). Within the UK, the range of *P. aegeria* has changed dynamically in the past 200 years. At the end of the 19th century there was a contraction of populations to south-west England, Wales and a refugial population in western Scotland, assumed to be in response to a change in climate (*Emmet & Heath, 1990*; *Warren et al., 2001*). Since the 1970s, the distribution of *P. aegeria* began to expand northwards from south-west England and Wales, argued to be mainly driven by increasing temperatures, but also other factors such as habitat preference. A secondary range expansion from the refugial population in western Scotland has recolonised much of central and northern Scotland. Increased habitat fragmentation of woodland habitats resulted in a greater than expected lag in the rate of range expansion compared to the climatic envelope (*Hill, Thomas & Huntley, 1999*). *P. aegeria* is now widespread across the UK.

A total of 774 male *P. aegeria* were hand netted across 54 sites (10–20 males/site) during the summers of 2016–2018 in the UK (Fig. 1; Table S1). In order to capture the pattern of range expansion, the site locations were chosen at a 10km grid scale aimed, firstly, at covering the whole of the geographic range and, secondly, to include the full range of number of years since *P. aegeria* was first recorded at 10km grid resolution. Specimens collected in 2018 were frozen on or the day after collection using a liquid nitrogen dry shipper and subsequently transferred to −80 ° C freezer for storage. Samples collected in 2016/17 were kept alive in a cool box for two days until transfer to a −80 ° C freezer. Permission for sampling at sites was obtained from landowners, including, but not limited to: Natural England, National Trust, Forestry Commission (England and Scotland), Woodland Trust, Yorkshire Wildlife Trust, Norfolk Wildlife Trust, local councils and site rangers through correspondence and verbal communication.

### Photography

Wings were carefully removed from the body at their point of attachment using fine scissors. The photographic setup consisted of Nikon D80 with Micro-Nikkor 40 mm lens attached to a camera stand, and two stand-alone speedlight (YN560IV) flashes (ISO 160, aperture f/25, exposure time of 1/80 s and compensation level +5.0). Size and shape analyses were carried out on *jpeg* files, but for colour analysis, to retain more information, images were converted from RAW Nikon images (NEF) to portable network graphics (png) using the programme XnConvert v.1.82. Photographs were calibrated by the R package Patternize (*Van Belleghem et al., 2018*), using the ColorGauge Micro Target (Image Science Associates) to account for any changes in lighting (Fig. 2A). Wings were graded into four wear categories, both
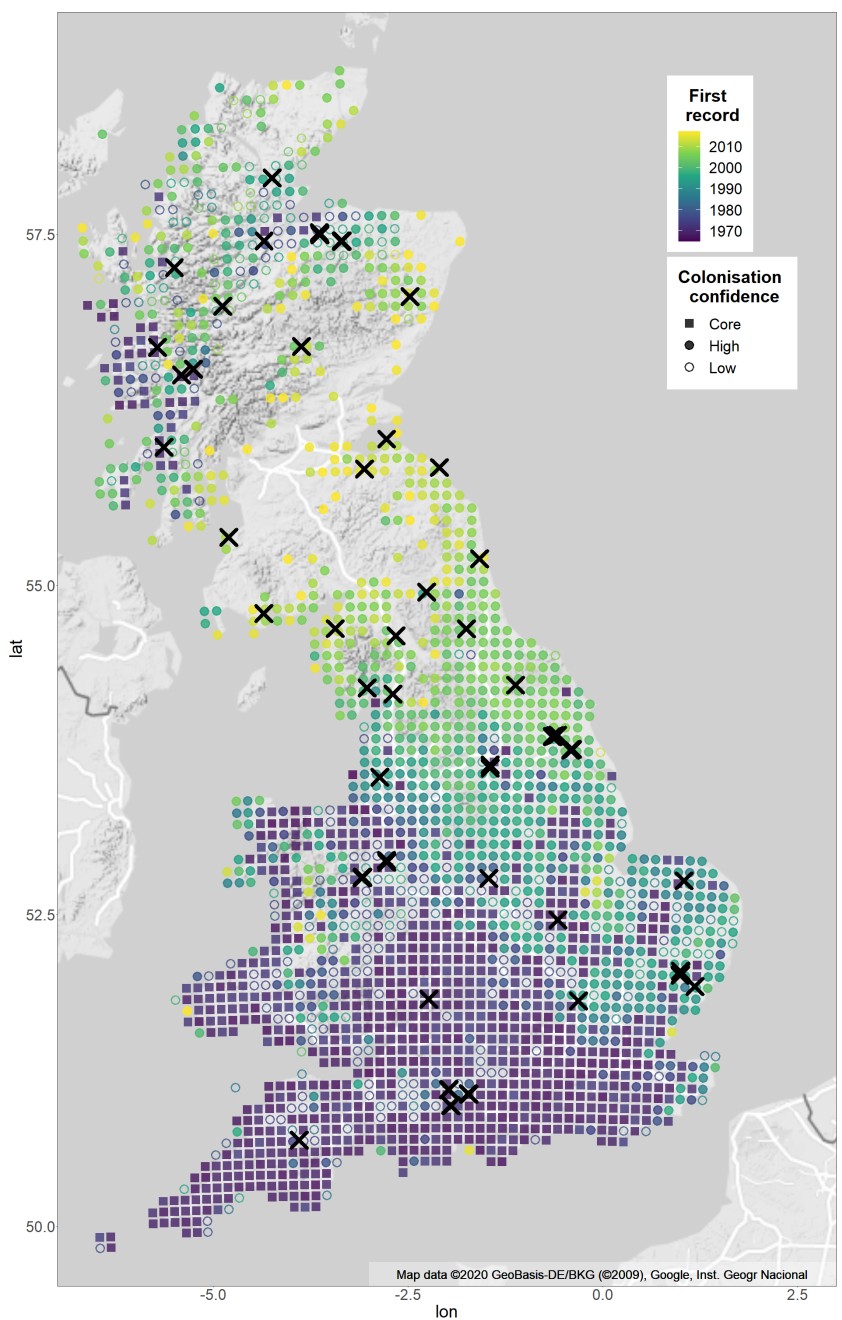

**Figure 1** **Sample sites and expansion of *Pararge aegeria* from 1965–2017 with confidence of colonisation year.** The map shows the distribution of *P. aegeria* and the pattern of range expansion from 1965–2017. Colours show years when *P. aegeria* was first recorded at a 10 km grid resolution. Crosses indicate site locations sampled for the study (Table S1 for details). The shape of each 10 km grid shows the confidence of the colonisation year record: grids colonised pre-1975 are considered core (squares); grids colonised after three or more well-recorded years (as defined by species richness records) are considered high confidence colonisations (filled circles); grids where the colonisation record is before the third year of good records are considered uncertain colonisations (unfilled circles). Two expansions have taken place, one from south-western England and the second from a refugial population in western Scotland. Map is from Google Static Maps API and plotted with ggmap in R (*Kahle & Wickham, 2013*).

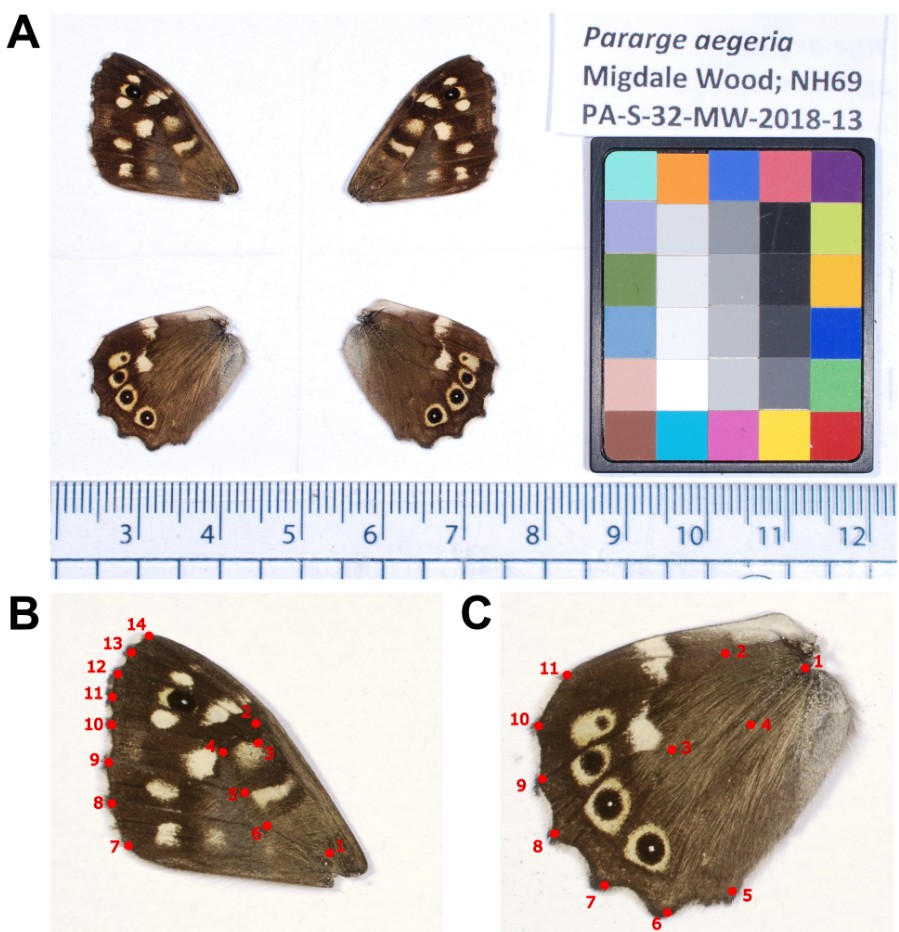

**Figure 2** *Pararge aegeria* **photographic set-up and landmarks.** Example image of dissected *P. aegeria* wings with the ColorGauge Micro Target (A). Forewing (B) and hindwing (C) landmarks. Each landmark was placed at either a vein-vein or vein-wing margin intersection.

for physical damage and scale damage (1= no damage, 4= significant/total damage). The sum of these two factors, assumed to be loosely correlated with butterfly age, was used as a factor in the analyses, or as a filter to remove significantly damaged and old butterflies. The left wings were selected as a priority for analysis, but if significant damage was present the right side was used instead.

## Environmental and demographic variables

We focused on four demographic or environmental variables: (1) number of years since first record or colonisation (years colonised); (2) latitude; (3) mean temperature during development; and (4) 10-year mean annual temperature.

To determine if developmental temperature accounts for any differences in morphology, latitudinal variation in emergence peaks had to be taken into account, due to *P. aegeria* being a multivoltine species. The specimens in this study were mainly collected late May—early August, which generally corresponds to the start of the second generation. However, more

southerly populations emerge earlier in the summer and undergo more generations than those further north. Therefore, the emergence patterns of each 10km grid sampled, based on 10-year abundance data from the Butterflies for the New Millennium (BNM) recording scheme (2007–2017), was investigated using generalised additive models, in the R package mgcv (*Wood, 2004*). This allowed identification of the number of generations at each grid, and the month of the second-generation peak. To assess the effect of temperature during embryo, larva and pupa development on morphological traits, we used the average temperature of the local second-generation peak month and the preceding two months, in the year of collection (e.g., for a site with a second generation peak in June, the mean temperature during development was calculated as the mean of April- June in the collection year).

To investigate potential morphological responses to multi-generational selection related to geographic variation in temperature, we used 10-year mean annual temperature ($T_{10}$), at 1km resolution, from the year prior to collection (e.g., for a butterfly sample collected in 2018, $T_{10}$ was calculated from 2007–2017). Ten years was chosen as a suitable timescale for phenotypic response to selection, even for more recently colonised populations, whilst reducing the effect of any large fluctuations in annual temperature. Temperature (monthly mean at a 1km resolution) data for 2007–2017 were obtained from the Met Office HadUK-Grid UKCP18 dataset, available through CEDA Archives. 2018 data were provided directly by the UK Met Office.

The pattern of range expansion was described by the number of years since the first record of *P. aegeria* at each 10km grid (referred to as years colonised), using distribution data from the BNM from 1965 onwards. The reliability of this type of data as an accurate reflection of changes in species distribution relies on recorder effort and geographic coverage of records across the UK. Previous studies that have accounted for recorder effort reveal that the expansion of *P. aegeria* is a true occurrence and not due to changes in recorder effort (*Parmesan et al., 1999*). To assess the reliability of the assumed pattern of expansion we applied an approach used in *Macgregor et al. (2019)*. For each grid, the percentage of regional species richness recorded was calculated (using data for 58 butterfly species provided by BNM, where regional species richness was the total number of species recorded in the 100 nearest neighbouring grids) in each year between 1965-2014. Grids were considered well-recorded in a given year if 10% or more of regional species richness was recorded. We then used this recording level to determine the level of confidence for the year each 10km grid was colonised by *P. aegeria*. Grids in which *P. aegeria* was recorded prior to 1975 were considered to be part of the core range of the species. From the remaining grids, we considered that we had high confidence of the colonisation year for grids which had been well-recorded in at least three years prior to the first record of *P. aegeria* (i.e., probable absence, followed by presence). We considered that we had low confidence of the colonisation year for grids in which the first record of *P. aegeria* coincided with or preceded the onset of good recording (i.e. *P. aegeria* first recorded in or before the third well-recorded year), since it was unclear whether such records represented true colonisation or simply the discovery of pre-existing populations. The reported pattern of range expansion is evident even when core and high confidence grids are considered in

isolation (Fig. 1), showing that it is not an artifact of increased or changing patterns of recorder effort. Furthermore, the majority of sites sampled are from 10km grids that are categorised as core or high confidence sites, supporting our use of 'years colonised' as an accurate metric for the sites studied.

## Wing landmarking and morphometrics

To allow for comparison of geometric morphometry and pattern across individuals, 14 and 11 landmarks were digitised on the forewing and hindwing, respectively, using tpsUtil version 1.78 (*Rohlf, 2019*) and tpsDig2 version 2.31 (*Rohlf, 2017*). All landmarks were placed on vein-vein or vein-wing margin intersections and provided adequate coverage of the overall shape and internal structure of the wings (Fig. 2). Landmarks were superimposed using generalised Procrustes analysis performed using the R package geomorph (*Adams & Otárola-Castillo, 2013*) and within MorphoJ version 1.07a (*Klingenberg, 2011*). This method standardises specimens to a common coordinate system through controlling size, orientation and position to align corresponding landmarks as closely as possible (*Rohlf & Slice, 1990*). Centroid size (CS) was also calculated from the landmarks (square root of the sum of squared distances between each landmark and the wing centroid), and was used as a measure of wing size and to account for allometry in the analysis of wing shape.

### Size analysis

Linear mixed effect models (LMM) fitted by restricted maximum likelihood (lme4 v.1.1-21 package; *Bates et al., 2015*), with *bobyqa* optimisation, was used to assess the effect of the four environmental factors on wing size, quantified as CS. Random effects included were: (1) site (10km grid) nested within regional expansion (i.e., south-west England or western Scotland), to account for variation amongst sites and expansions, and (2) number of Julian days before or after the peak of the second generation at a 10km scale that the sample was collected on (referred to as the standardised collection date). Correlation of explanatory variables was investigated in R with scatterplots and the Pearson correlation coefficient. The effect of voltinism on wing size was tested using a two-sample $t$-test.

Significance testing for LMMs is not straightforward, as the denominator for degrees of freedom is difficult to obtain for models with multiple levels (*Baayen, Davidson & Bates, 2008*); therefore the package lme4 does not produce $p$-values (*Bates et al., 2015*). Instead, the $t$-value from the LMM indicates the strength of the effect and some authors suggest a $t$-value of magnitude over 1.96 can be considered significant, following the $t$-as-$z$ approach (*Luke, 2017*). Therefore, traditional $p$-values are not presented for LMMs in this study, and a $t$-value with a magnitude of 1.96 or above is considered significant.

### Shape analysis

To investigate shape changes, independently of size, a multivariate regression of the Prosecutes coordinates against wing size (log CS) was carried out (10,000 permutations). This method accounts for allometric patterns by producing a regression score that corresponds to the shape variable with the greatest covariation to size. The residuals of this regression can therefore be treated as a size-adjusted shape variable. This method has been used widely to account for allometry in many morphometric studies (*Van Heteren et al.,*

*2016*; *Curth, Fischer & Kupczik, 2017*). An analysis of covariance (ANCOVA) was used to determine if allometry was significantly different between forewings and hindwings.

The resulting size-adjusted variables were then used for a two-block partial least squares (2B-PLS) method across all individuals (*Rohlf & Corti, 2000*). The 2B-PLS method aims to capture the greatest amount of covariation between two blocks of variables (here size-adjusted shape as block one, and environmental variables as block two) of equal weight. This method calculates a RV coefficient, that can be interpreted as a multivariate generalization of the bivariate $R^2$, and used to determine the strength of the covariation between blocks (*Klingenberg, 2009*). A permutation test (10,000 repetitions) was used to compare the observed association against the null hypothesis of complete independence. Shape changes associated with PLS axes are shown using wireframe diagrams against the mean (or consensus) wing shape. All 2B-PLS analyses were carried out in MorphoJ v1.07a (*Klingenberg, 2011*) and plotted with ggplot2 (*Wickham, 2016*).

## Wing colour, pattern and contrast

*Pararge aegeria* wings are brown with a principally cream spot pattern. Four complementary measures of colour and pattern were investigated: (1) average degree of lightness across the basal 3rd and entire wing surfaces (dorsal and ventral surfaces on forewing and hindwing); (2) average lightness within brown and cream areas (forewing dorsal only); (3) the relative area of brown and cream (forewing dorsal only); and (4) the contrast between brown to cream areas (forewing dorsal only).

Degree of lightness of the forewing and hindwing dorsal and ventral surfaces was investigated using the mean grey value for the whole wing and the basal third of the wing, closest to the body (basal colour). It has been suggested that wing basal colour is the most important for thermal regulation (*Wasserthal, 1975*). An ImageJ (FiJi distribution) macro was created to select the individual wings from the background, rotate them to the same orientation, separate wings into thirds (perpendicular to the longest axis), convert RGB images to 8-bit grey, and calculate the mean grey value for the whole wing and each wing section. The full 8-bit grey scale ranges from 0 (complete black) to 255 (complete white). Wings with scale damage of 4 were removed from the analysis, leaving 709 forewings and 642 hindwings. The effect of the environmental variables on mean grey values was analysed in a LMM, fitted by restricted maximum likelihood and *nmkbw* optimisation (lme4 v.1.1-21 package; *Bates et al., 2015*). Site nested within regional expansion (i.e., south-west England or western Scotland), and the standardised collection date metric were included as random effects. Significance testing followed the method detailed for size analysis.

To investigate variation in the brown and cream areas separately (dorsal forewing only as it has the most discrete pattern), a macro script for ImageJ (FiJi distribution) was written to calculate the mean grey value and area (number of pixels) for each (filtered for scale damage of 4). Linear regression was used to assess the relationship of brown (or cream) area with latitude, and an ANCOVA to determine if these relationships differed significantly from each other. The relationship of brown to cream colours in the forewing was quantified by linear regression on the residuals of each colour to latitude (to focus on the underlying relationship). Finally, the difference between the cream and brown mean grey value was
calculated to produce a simple measure of average contrast between the dark and light areas of the wing. The effect of the environmental variables on contrast between brown and cream was analysed in a LMM, fitted by restricted maximum likelihood and *nmkbw* optimisation (lme4 v.1.1-21 package; *Bates et al., 2015*). The random effects included were the same as for the overall lightness analysis.

## RESULTS

### Wing morphometrics
#### *Size*
Forewing and hindwing size increase significantly with latitude ($t$-values $> 1.96$) and in more recently colonised populations (shown by negative relationship of size to an increase in number of years colonised), consistent with Bergmann's rule and spatial selection. Forewing size is also weakly associated with temperature during development ($t = 2.64$) but not in hindwings ($t = 1.65$). In general, each of the environmental factors (latitude, years colonised, 10-year temperature average, and temperature during development) show a consistent effect (both in the strength and direction) on forewing and hindwing size, although latitude seems to have a stronger effect on hindwings compared to forewings (Table 1; Fig. 3). $T_{10}$ (10-year mean annual temperature) produced the lowest t-values (0.57, 0.58) across the environmental variables for both wings, indicating no effect of recent past temperature on wing size. Correlation between explanatory variables was considered acceptable for LMMs (Fig. S1). The Pearson correlation coefficient ranged between 0.14 (years colonised with temperature during development) to $-0.74$ (latitude with mean annual temperature). Although latitude is often considered a proxy for annual temperature, it also incorporates other environmental gradients that follow latitude, for example day length and amount of sunlight etc. Therefore, it was decided to retain both $T_{10}$ and latitude in the model.

The populations varied between two and three generations per year across the 10km grid sampled, and size of forewings (mean CS of 2.60 (2 generations) and 2.55 (3 generations)) and hindwings (mean CS of 2.39 (2 generations) and 2.33 (3 generations)) are significantly smaller (1.9% and 2.5% smaller in forewings and hindwings, respectively) in populations with three generations (t-test: $t = 4.96$, $df = 661.36$, $p < 0.001$ for forewing; $t = 4.45$, $df = 598.26$, $p < 0.001$ for hindwing).

#### *Shape*
Shape is significantly associated with size (log CS) both for forewing and hindwing, reflecting shape allometry ($p < 0.0001$; Figs. 4A and 4B). Allometry is less pronounced in the forewing, where the regression explained 1.25% of shape variance compared to 4.14% in the hindwing (ANCOVA of regressions, $F = 21.53$, $df = 1$, $p < 0.001$; slope of $1.31 \pm 0.14$ and $2.25 \pm 0.14$ respectively; Figs. 4A and 4B). Shape changes for forewings and hindwings relative to size and the consensus shape are shown in Figs. 4C–4F, respectively. Overall, larger wings have increased width and roundness compared to smaller individuals. The shape difference between small and large forewings is most noticeable for landmark 7, which moves further away from the consensus shape with increasing size, and for

**Table 1 Linear mixed model results for forewing and hindwing size (centroid size).**

| Predictors | Forewing centroid size | | | | Hindwing centroid size | | | |
|---|---|---|---|---|---|---|---|---|
| | Est. | SE | df | t | Est. | SE | df | t |
| (Intercept) | 0.31 | 0.56 | 20.44 | 0.55 | −0.69 | 0.54 | 1.62 | −1.28 |
| Latitude | 0.04 | 0.01 | 13.31 | **3.85** | 0.05 | 0.01 | 0.48 | **6.43** |
| Years colonised | −0.002 | 0.001 | 15.09 | **−2.83** | −0.002 | 0.00 | 0.77 | **−2.53** |
| Mean 10yr annual temp ($T_{10}$) | 0.01 | 0.02 | 40.83 | 0.57 | 0.012 | 0.02 | 28.63 | 0.58 |
| Temp. during development | 0.02 | 0.01 | 31.30 | **2.64** | 0.02 | 0.01 | 33.83 | 1.65 |
| **Random Effects** | | | | | | | | |
| $\sigma^2$ | 0.007 | | | | 0.007 | | | |
| $\tau_{00}$ | 0.001 Julian day difference | | | | 0.003 Julian day difference | | | |
| | 0.002 Grid.10km:Expansion | | | | 0.002 Grid.10km:Expansion | | | |
| | 0.003 Expansion | | | | 0.000 Expansion | | | |
| ICC | 0.47 | | | | 0.42 | | | |
| N | 40 Grid.10km:Expansion | | | | 40 Grid.10km:Expansion | | | |
| | 2 Expansion | | | | 2 Expansion | | | |
| | 51 Julian day difference | | | | 51 Julian day difference | | | |
| Observations | 686 | | | | 614 | | | |
| Marginal $R^2$ / Conditional $R^2$ | 0.33 / 0.64 | | | | 0.48 / 0.70 | | | |

**Notes.**
Significant *t* values with a magnitude greater than 1.96 are indicated in bold.
$\sigma^2$, Residual variance; $T_{00}$, Random effect variance; ICC, Interclass correlation coefficient.

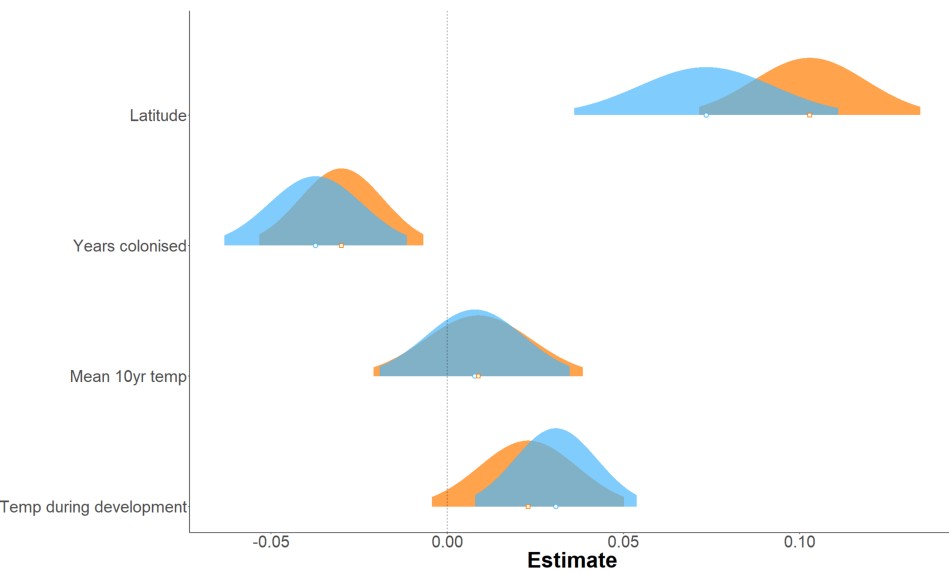

**Figure 3 Effect of environmental variables on forewing and hindwing size (centroid size).** Linear mixed model estimated of the effect, shown as probability distributions, of four environmental variables on forewing (blue) and hindwing (orange) centroid size of *P. aegeria*.

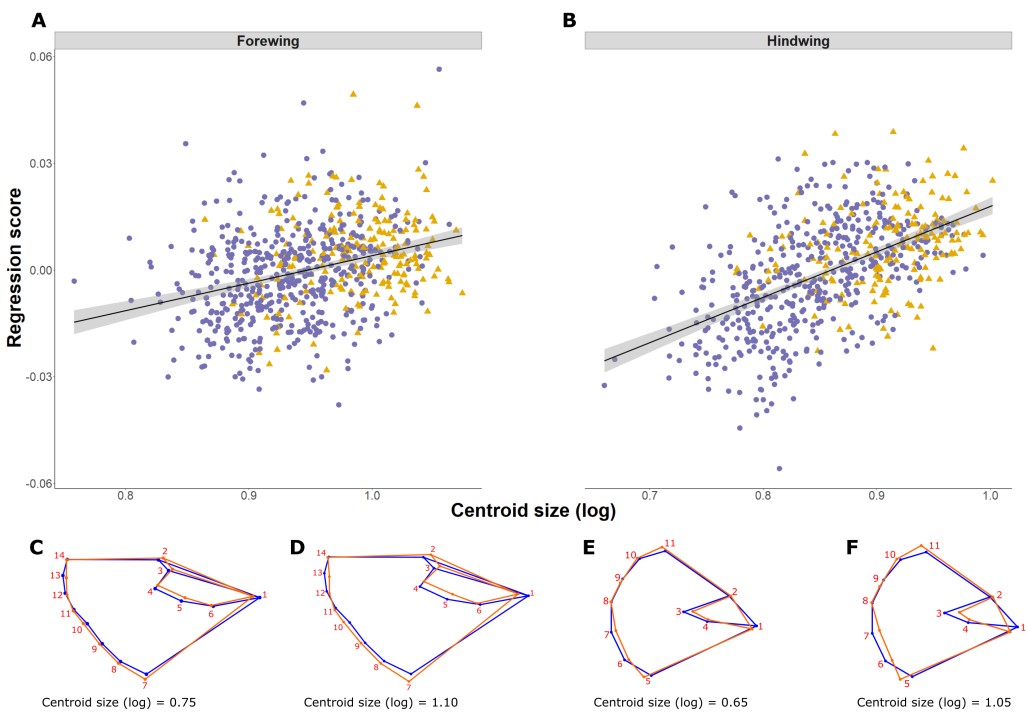

**Figure 4** *Pararge aegeria* **wing allometry.** Regression (solid line with standard error in grey) of the wing shape variable on forewing (A) and hindwing (B) size (log centroid size, CS) in *Pararge aegeria*. Data are separated into the two regional populations (England, purple circles; Scotland, yellow triangles). Shape changes associated with different values of log CS are shown using a wireframe diagram for forewings (C and D) and hindwings (E and F). The blue wireframe is the consensus (or average) shape of all the individuals, whilst the orange indicates the shape at each log CS as indicated.

landmark 1, which is shifted inwards, producing relatively broader and shorter wings. In the hindwings, landmarks 5 and 11 are more separated from each other in large hindwings, resulting in relatively wider wings compared to smaller hindwings.

The 2B-PLS analysis focused on covariation between the size-adjusted shape (block 1) and environmental variables (block 2). Overall, the permutation test showed a significant covariation between the two blocks both for the forewing ($p = 0.006$) and hindwing ($p = 0.0001$), supporting non-independence of the two blocks. The overall strength of association between the two blocks (as explained by the RV coefficient) is weak, at 0.012 and 0.025, respectively. The first PLS axis (PLS1) explained 73% and 91% for forewing and hindwing, respectively, and was principally loaded by years colonised and latitude (for both wings), but showed a weak correlation (0.20 and 0.22 respectively; Table 2; Figs. 5A and 5B). This suggests that, out of the environmental factors studied, the range expansion process has the largest effect on shape. $T_{10}$ was always loaded on the fourth PLS axis, explaining the least variation across both wings, implying a minimal effect of recent past temperature on wing shape.

Shape changes associated with PLS1 (associated most strongly with years colonised; Figs. 5C–5F) and the second PLS axis (PLS2; mainly latitude; not shown) indicate a general

**Table 2 Statistics for two-block partial least squares (2B-PLS) anlaysis for size-adjusted shape variables to environmental variables for forewing and hindwings.** The environmental variable most strongly associated with each partial least square (PLS) axis is indicated with an asterisk. Significant correlations (*p*-values) using 10,000 permutations are indicated in bold.

| | FOREWING | | | | HINDWING | | | |
|---|---|---|---|---|---|---|---|---|
| | PLS1 | PLS2 | PLS3 | PLS4 | PLS1 | PLS2 | PLS3 | PLS4 |
| *Loadings against:* | | | | | | | | |
| Latitude | −0.18 | −0.77* | 0.55 | 0.28 | −0.16 | −0.88* | 0.20 | −0.40 |
| Years colonised | 0.98* | −0.14 | 0.12 | 0.00 | 0.98* | −0.18 | 0.05 | 0.02 |
| Dev. temp. | −0.02 | 0.58 | 0.81* | 0.00 | −0.02 | 0.17 | 0.98* | 0.12 |
| Mean 10yr temp ($T_{10}$) | 0.05 | 0.22 | −0.17 | 0.96* | 0.09 | 0.41 | 0.04 | −0.91* |
| | | | | | | | | |
| Covariation explained | 72.96 | 23.60 | 2.73 | 0.71 | 90.92 | 7.41 | 1.5 | 0.14 |
| Correlation between blocks (r) | 0.20 | 0.25 | 0.18 | 0.19 | 0.22 | 0.26 | 0.18 | 0.15 |
| Correlation coeff. *P*-value (perm.) | **0.001** | **<.0001** | **0.001** | **0.0001** | **0.0002** | **<.0001** | **0.0012** | 0.02 |

tendency towards longer, narrower forewings (i.e., with increasing numbers of years colonised). Within PLS1 this increase mainly occurs between the base or shoulder of the wing (landmark 1) to the apex (landmarks 12-14), while the distance between landmark 1 and 7 decreases (Figs. 5C and 5D). In comparison, the hindwing PLS1 (and PLS2) becomes more rounded. For PLS1, the increased roundness seems to be due to the majority of landmarks at the tail edge of the wing being more separated from one another (Figs. 5E and 5F).

## Colour, pattern and contrast

The average lightness (average grey value) of whole wing surfaces, whether dorsal or ventral, has a complex relationship with latitude, with periodic fluctuations of lightness that are consistent in the forewing and hindwing (Fig. 6). The relationship of average lightness with the demographic and environmental variables are similar across the basal third of the wing (Table 3) and whole wing (Table S2). The lightness of ventral surfaces, both of forewings and hindwings, becomes significantly darker with increasing temperature during development, but significantly lighter with increasing latitude (Table 3; greyness value of 0 is black and 255 is white). The direction of effects is consistent among all four wing surfaces, although the trend is not significant for dorsal surfaces. There is no detectable effect of $T_{10}$ and years colonised on any wing surfaces. These effects account for variation due to population (England or Scotland) and sampling date.

The relationship between latitude and mean grey value of the dorsal forewing depends significantly on the colour considered (ANCOVA, $F = 27.69$, $df = 1$, $p < 0.001$). Both brown and cream become significantly lighter with latitude ($p < 0.001$), but the cream area becomes lighter faster (i.e., further north) than the brown area (slopes of $3.38 \pm 0.25$ vs $1.49 \pm 0.26$ greyscale units per degree of latitude, respectively). These slopes reflect a strong positive correlation between the lightness of brown and cream areas, which is independent of the latitudinal trend ($R^2 = 0.68$; $p < 0.001$; Fig. 7). However, the relative proportion of the dorsal forewing surface that is brown increases significantly with latitude ($R^2 = 0.09$, $df = 702$, $p < 0.001$). The analysis of contrast (difference between cream and brown mean
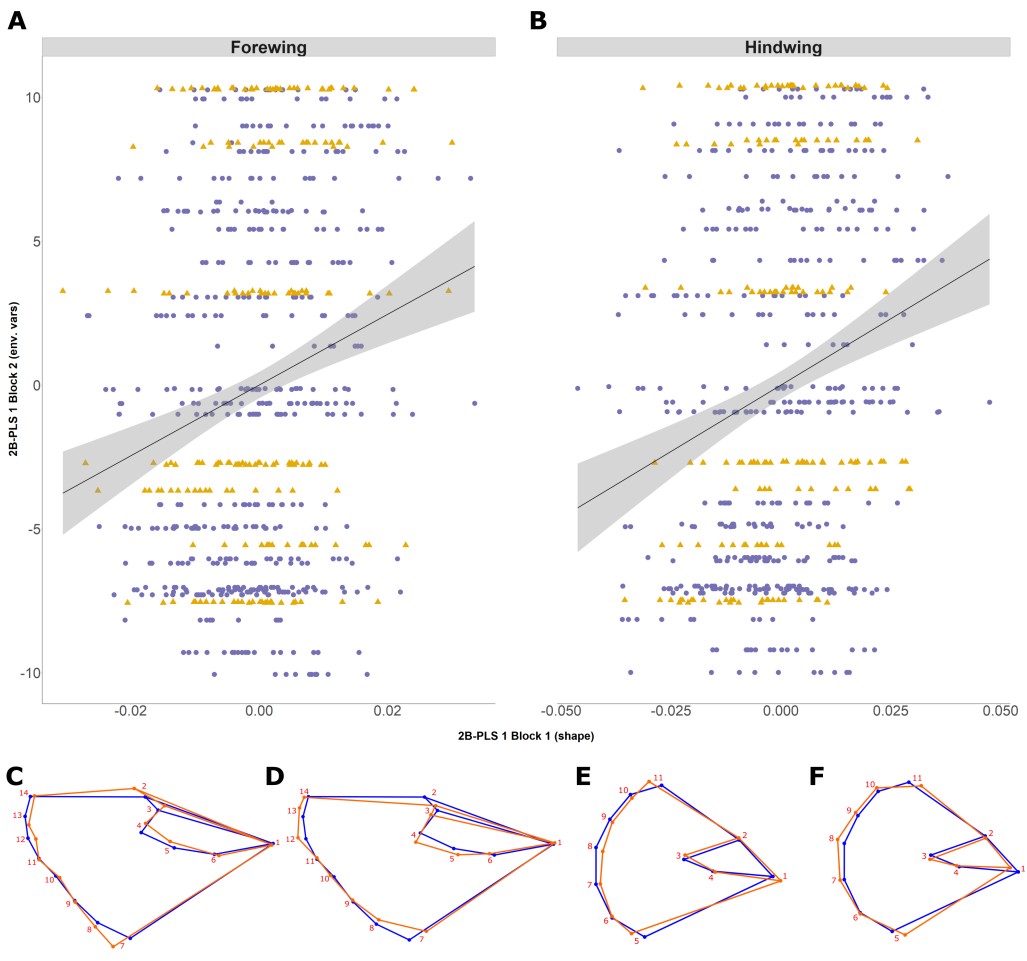

**Figure 5 Two-block partial least squares (2B-PLS) axis 1 shape changes for forewing and hindwing.** The plot shows PLS1 axis for block 1 (i.e., size-adjusted shape variable) against PLS1 axis for block 2 (environmental variables) for forewing (A) and hindwing (B). The points are coloured according to which expansion they originate (England, purple circles; Scotland, yellow triangles). The change of shape associated across the PLS1 axis is shown using wireframe plots, where the blue line is the consensus shape (or average) and orange is the shape present at the axis value of −0.1 (C and E) and 0.1 (D and F) for forewing and hindwings respectively.

grey values) shows that the level of contrast increases significantly with latitude (*est.* = 1.26 ± 0.50, *df* = 46.37, *t* = 2.54; Fig. 8), even when accounting for population (England or Scotland) variation and collection date relative to the site-specific emergence peak.

## DISCUSSION

This study documents detailed wing morphological variation (size, shape and colour) in the Speckled Wood butterfly, *P. aegeria*, across two recently expanded populations in mainland Britain, suggesting differing responses to environmental and demographic factors. The size of *P. aegeria* increases with latitude, consistent with Bergmann's rule, and during the range expansion process, with more recently colonised populations being larger
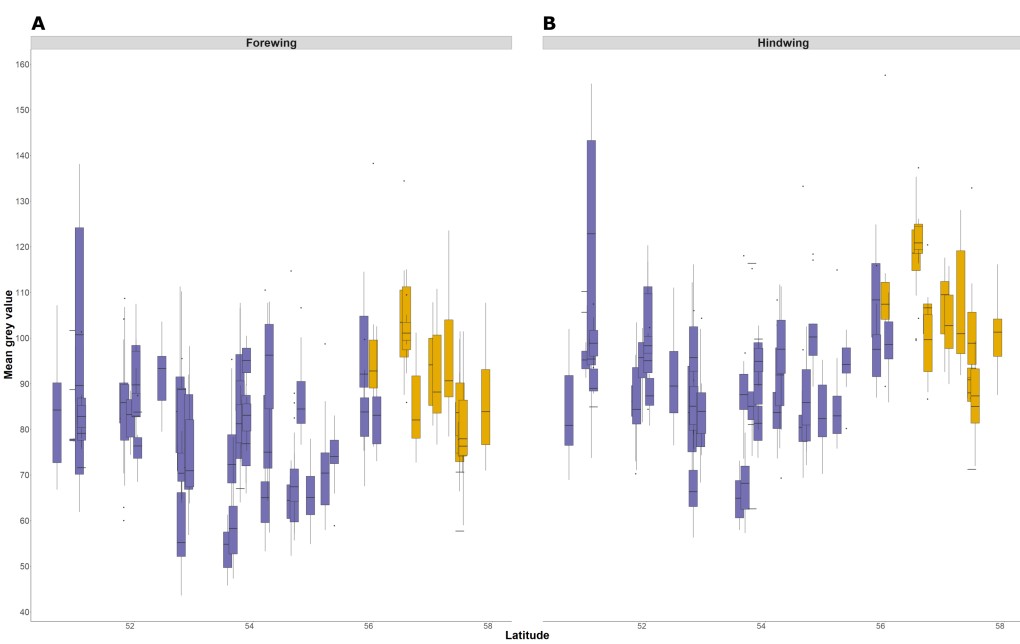

**Figure 6 Forewing and hindwing whole wing dorsal lightness with latitude.** The forewing (A) and hindwing (B) dorsal lightness (mean grey value) relationship with latitude. A mean grey value of 255 is white and 0 is black. Populations are coloured according to England (purple) or Scotland (yellow) expansion.

than core populations. Shape changes, independent of size, are most strongly associated with colonisation history. Forewing shape becomes more rounded, whereas hindwing shape becomes longer, in more recently colonised populations and with latitude. The distribution of average lightness (opposite of melanism) is more strongly associated with temperature during development than it is to latitude, and runs contrary to the traditional thermal melanism hypothesis. Furthermore, the area of brown relative to cream increases with latitude, but not enough to overcome the general lightening in both areas. Finally, the contrast between brown and cream areas increases with latitude, accounting for the human perception that individuals become darker further north. Overall, this study sheds light on the interaction of temperature-sensitive plastic traits and selection during a mainly climate-driven range expansion.

During range expansion, sections of a population experience different and new environmental conditions that may result in local adaptation, be it through genetic changes or phenotypic plasticity. Our analyses are of wild-caught individuals who experienced different environments during development, making it impossible to separate plastic from genetic effects. Previous broad sense heritability estimates in a *P. aegeria* population from southern Sweden ($h^2 = 0.38$–$0.45$) indicate high potential for evolutionary responses in comparable morphological traits, including wing size and colour pattern (*Van Dyck & Matthysen, 1998*). Furthermore, a positive correlation between thorax investment and wing shape, which was associated with acceleration performance during take-off in males, had a heritability of 0.15 (*Berwaerts, Matthysen & Dyck, 2008*). This heritability estimate

Peerj

**Table 3  Linear mixed model results for forewing and hindwing basal mean grey value.**

| Predictors | FOREWING | | | | | | | | HINDWING | | | | | | | |
|---|---|---|---|---|---|---|---|---|---|---|---|---|---|---|---|---|
| | Dorsal | | | | Ventral | | | | Dorsal | | | | Ventral | | | |
| | *Est.* | *SE* | *df* | *t* | *Est.* | *SE* | *df* | *t* | *Est.* | *SE* | *df* | *t* | *Est.* | *SE* | *df* | *t* |
| (Intercept) | −1.16 | 92.93 | 25.07 | −0.01 | 36.95 | 76.42 | 54.41 | 0.48 | 82.14 | 112.92 | 28.97 | 0.73 | −8.18 | 86.91 | 55.81 | −0.09 |
| Latitude | 1.47 | 1.53 | 16.74 | 0.96 | 2.15 | 1.07 | 48.92 | **2.01** | 0.81 | 1.87 | 22.16 | 0.43 | 2.88 | 1.23 | 49.29 | **2.35** |
| Mean 10-year annual temperature ($T_{10}$) | 4.32 | 2.54 | 59.41 | 1.70 | 1.67 | 2.53 | 58.34 | 0.66 | 1.50 | 3.03 | 59.36 | 0.49 | 2.59 | 2.89 | 59.75 | 0.90 |
| Temperature during development | −2.34 | 1.33 | 38.34 | −1.76 | −3.27 | 1.41 | 38.74 | **−2.33** | −1.88 | 1.59 | 37.78 | −1.18 | −3.23 | 1.63 | 39.04 | **−1.98** |
| Years colonised | 0.07 | 0.14 | 18.79 | 0.50 | 0.18 | 0.10 | 38.35 | 1.82 | −0.03 | 0.16 | 24.65 | −0.18 | 0.13 | 0.12 | 38.65 | 1.08 |
| **Random Effects** | | | | | | | | | | | | | | | | |
| $\sigma^2$ | 82.56 | | | | 88.93 | | | | 101.88 | | | | 79.46 | | | |
| $\tau_{00}$ | 18.41 Julian day difference | | | | 12.15 Julian day difference | | | | 27.06 Julian day difference | | | | 18.77 Julian day difference | | | |
| | 70.45 Grid.10km:Expansion | | | | 86.82 Grid.10km:Expansion | | | | 101.35 Grid.10km:Expansion | | | | 11.79 Grid.10km:Expansion | | | |
| | 88.51 Expansion | | | | 0.0001 Expansion | | | | 172.05 Expansion | | | | 0.0002 Expansion | | | |
| ICC | 0.68 | | | | 0.53 | | | | 0.75 | | | | 0.63 | | | |
| N | 43 Grid.10km | | | | 43 Grid.10km | | | | 43 Grid.10km | | | | 43 Grid.10km | | | |
| | 2 Expansion | | | | 2 Expansion | | | | 2 Expansion | | | | 2 Expansion | | | |
| | 53 Julian day difference | | | | 53 Julian day difference | | | | 53 Julian day difference | | | | 53 Julian day difference | | | |
| Observations | 709 | | | | 709 | | | | 642 | | | | 641 | | | |
| Marginal $R^2$/ Conditional $R^2$ | 0.05 / 0.70 | | | | 0.20 / 0.62 | | | | 0.03 / 0.75 | | | | 0.20 / 0.71 | | | |

**Notes.**

Significant *t* values with a magnitude greater than 1.96 are indicated in bold.

$\sigma^2$, Residual variance; $T_{00}$, Random effect variance; ICC, Interclass correlation coefficient.

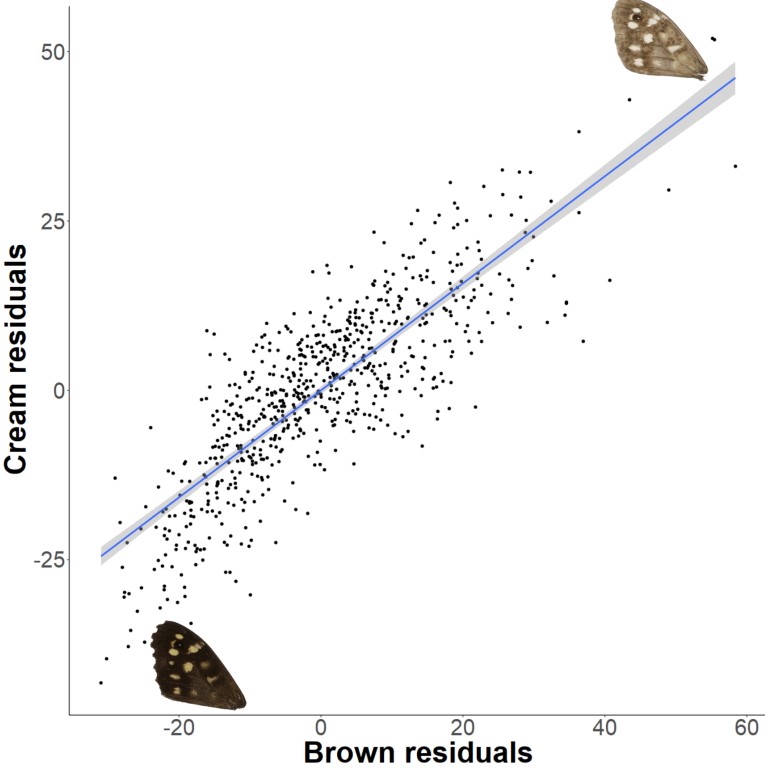

**Figure 7** **Forewing brown and cream lightness correlation independent of latitude.** There is a strong correlation of brown and cream lightness (mean grey value), even when accounting for latitude. Example wings are inserted to demonstrate the colour change occurring across the regression.

is specific to acceleration performance, and so caution should be taken if relating this to general dispersal ability. Thus, although we cannot quantify the effect of developmental environment on our phenotypic distributions, there is likely to be a degree of heritable genetic variance in all of our measured traits.

## Size and shape variation

Forewing and hindwing size in *P. aegeria* increases with colonisation and latitude. Larger individuals were found in more recently colonised populations, following the expectation under spatial selection if larger wings are associated with dispersal, for which there is some evidence (*Sekar, 2012*; *Flockhart et al., 2017*). Our findings support the conclusion of *Hughes, Hill & Dytham (2003)*, which were based on thorax size across a much more limited number of sites and geographic range of *P. aegeria*. On the reasonable assumption that wing size is directly correlated to body size in this species, as in other Lepidoptera (*Chai & Srygley, 1990*), increased wing size with latitude follows Bergmann's rule, in agreement with previous studies (*Dennis & Shreeve, 1989*; *Sibly, Winokur & Smith, 1997*). Temperature during development shows a positive relationship with forewing size, which runs counter to the prediction of the temperature size rule (*Atkinson & Sibly, 1997*), but is consistent with experimental results (C Macgregor, 2020, unpublished data). Within

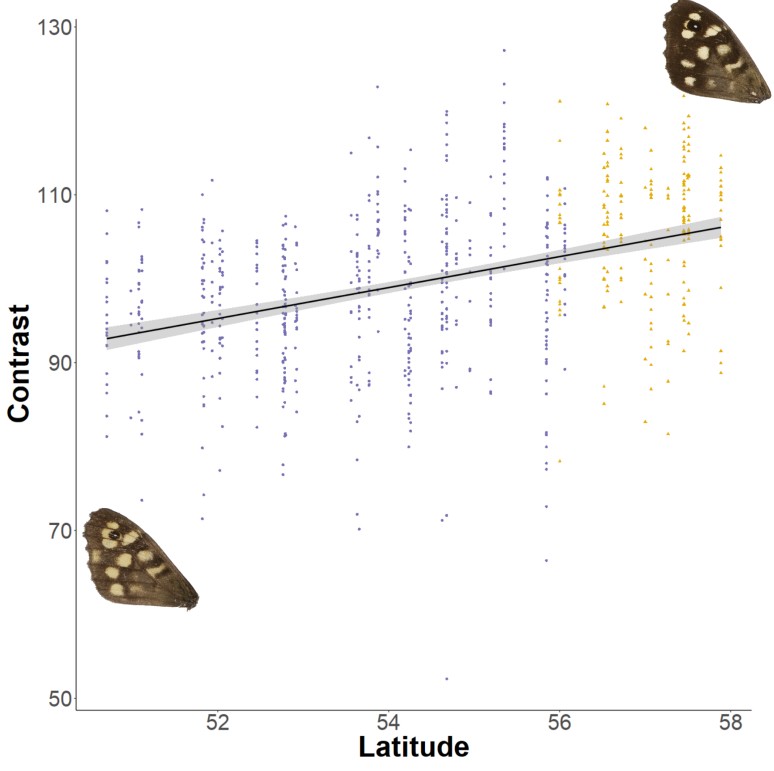

**Figure 8** **Effect of latitude on contrast between brown and cream areas.** The relationship between contrast (difference between mean grey values of the cream and brown areas) on the dorsal forewing surface and latitude. The points are coloured according to which expansion they originate (England, purple circles; Scotland, yellow triangles). Example wings are inserted to demonstrate the colour change occurring at contrasting ends of the relationship.

this underlying temperature-size relationship, there is also an additional effect of season length and development time, as reflected by the observed reduction in size with number of generations. A limitation of using ambient temperature as a measure of temperature during development is that the micro-climate that individuals experience can be significantly modified by behaviour and other environmental factors, such as humidity and sunlight. Under fast demographic change, as in the range expansion of *P. aegeria*, the process of spatial selection may override selection pressures from environmental gradients. For example, in the Scottish expansion, wing size has responded more strongly to selection for dispersal than to the environmental gradients (all recently colonised site are noticeably larger than the core populations, irrespective of latitude and temperature).

The shape of forewings and hindwings, independent of size, were found to change during the range expansion process and with increasing latitude. Forewings are more rounded and hindwings are narrower in more recently colonised populations. Spatial selection during a range expansion often results in increased dispersal ability towards the leading edge (*Phillips, Brown & Shine, 2010*). The finding that the number of years colonised has the strongest loading to shape in both the forewings and hindwings suggests these shapes

are more dispersive in this species. However, interpreting the functional consequences of fine-scale wing shape variation for different dimensions of flight performance (e.g., dispersal efficiency, acceleration, manoeuvrability) and tying shape changes to adaptive evolution is a complex and currently an unresolved problem (*Le Roy, Debat & Llaurens, 2019*). In fact, large, long and more pointed forewings are often considered the most dispersive form as this is associated with (less metabolically costly) gliding flight and migratory species, e.g., the Monarch butterfly (*Danaus plexippus*; *Altizer & Davis, 2010*; *Flockhart et al., 2017*). However, this assumption may not hold true for all species. For example, in the Glanville Fritillary (*Melitaea cinxia*), dispersive females have more rounded wings (*Breuker, Brakefield & Gibbs, 2007*), whilst another study found little or no evidence linking wing morphology to dispersal ability (*Hanski et al., 2002*).

In *P. aegeria*, the female is thought to be the more dispersive sex (*Shreeve, 1986*) and, in general, is larger with more rounded wings compared to males (*Pellegroms, Van Dongen & Van Dyck, 2009*). Mate location strategy in male *P. aegeria* varies between two behaviours, perching or patrolling, requiring different flight dynamics (*Shreeve, 1984*). Perchers require high acceleration to defend a territorial sunlit patch and intercept passing females, whereas patrollers require attributes for more sustained flight as they move from one spot to another in search of a female. The differing flight requirements of these behaviours is reflected in their thorax size, wing morphology and colour (*Van Dyck, Matthysen & Dhondt, 1997*; *Van Dyck & Matthysen, 1998*; *Berwaerts, Van Dyck & Aerts, 2002*). An increase in forewing roundness was also found in *P. aegeria* males across a large latitudinal (700km) gradient from France to Netherlands, where a decrease in aspect ratio (AR; calculated as 4 x forewing length/forewing area), was found further north (*Van dewoestijne & Van Dyck, 2011*). The AR is widely used as a predictor of flight performance but does not account for allometry, as in the geometric morphometric approach used in this study. Therefore, comparisons between studies based on AR and this one should be considered carefully. Nonetheless, Vandewoestijne and Van Dyck's finding supports the pattern seen here, suggesting a common trend in wing shape with increasing latitude in this species.

The majority of studies focus solely on forewing shape. Although butterflies are mostly antermotoric (require, and predominantly use, forewings for flight), hindwings increase linear and turning acceleration, so are particularly important for aerial agility and predator evasion (*Jantzen & Eisner, 2008*). The distinct roles of wings in flight, coupled with our finding that shape changes are different between forewings and hindwings, suggest that evolutionary factors may differentially affect forewing and hindwing shape. This study highlights the need to include both wings in future studies.

## Colour and pattern variation

*Pararge aegeria* has been previously described as becoming darker further north in Britain (*Dennis & Shreeve, 1989*). However, using quantitative measurements and a more spatially extensive sample, our results do not follow the expectation from the thermal melansim hypothesis (i.e., decreasing lightness with latitude). Lightness fluctuates with latitude producing a wave-like pattern that is consistent across both wings. The basal area becomes lighter with latitude, whereas average lightness decreases with temperature

during development (darker with increasing temperature). Increased lightness at cooler temperatures during development could support an energetic trade-off between overall growth and melanin production, which is costly to synthesise (*Talloen, Van Dyck & Lens, 2004*). The production of melanin may be subject to selection pressures unrelated to thermoregulation (*True, 2003*). The level of melanism has implications for disease resistance (*Wilson et al., 2001*; *Dubovskiy et al., 2013*) UV protection (*Bishop et al., 2016*; *Katoh, Tatsuta & Tsuji, 2018*), predation (*Bond & Kamil, 2002*) and sexual selection (*Jiggins et al., 2001*; *Kemp, 2007*). Our samples are of male individuals and so changes in colour due to selection for thermoregulatory properties (increased melanism) are likely to be constrained by sexual selection (*Tuomaala, Kaitala & Rutowski, 2012*). Behavioural traits such as posture during basking, which was not assessed in this study, also effect thermal regulation (*Kingsolver, 1985*; *Berwaerts et al., 2001*). In *P. aegeria*, wing colour is associated with alternative mate-location strategies, with perchers being lighter in colour than patrollers (*Van Dyck & Matthysen, 1998*). The sample used here is likely biased towards perchers due to the increased likelihood of spotting and netting perching individuals. Furthermore, previous studies that find decreased lightness with latitude also include thorax colour which was not possible in this study (*Zeuss et al., 2014*).

The perceived darkening of *P. aegeria* at higher latitudes (as reported by recorders and the authors of this study) is probably due to the relative changes of brown and cream areas on the forewing. The area of brown colour increases significantly with latitude, which may also have thermal regulatory consequences. The strong correlation between the lightness of brown and cream areas is indicative of an underlying genetic and developmental mechanism controlling the 'background' production of melanin across the whole wing surface that is also sensitive to environmental cues during the larval or pupal stages. Nevertheless, the cream colour increases in lightness significantly faster than the brown colour, resulting in an overall increase in contrast between the brown and cream patches. Few studies have looked at the effect of pattern on thermoregulation, but the wing band pattern in Banded Peacock (*Anartia fatima*) has been shown to slow the rate of heating but not the overall thoracic temperature equilibrium (*Brashears, Aiello & Seymoure, 2016*). The consequences of the specific traits detailed here on thermal properties of *P. aegeria* wings have not been studied to date, and so conclusions relating these findings to the thermal regulation should be made with care.

## CONCLUSION

We have shown that the rapid expansion of *P. aegeria* across a temperature gradient in a spatially fragmented landscape is associated with shifts in morphological traits that are differentially affected by the environmental and demographic factors studied. Wing size and shape are most strongly linked to latitude, following Bergmann's rule, and colonisation, consistent with selection on dispersal. The spatial distribution of average lightness is only weakly related to latitude and more associated with a plastic response to temperature during development, which on the surface would appear to run contrary to the thermal melanism hypothesis. Interpretation of the patterns observed here must take account of

the dynamic nature of this recent and ongoing range expansion. Populations are likely to be changing phenotypically through adaptation to local environmental conditions and secondary immigration. Genetic drift associated with colonisation, and evolutionary time-lags, may also account for some of the high variance in phenotype-environment associations. The non-equilibrium state of many of the local populations sampled, and an overriding role of selection for traits linked directly to range expansion, may explain some of the weak phenotype-environment associations observed. The planned application of genetic markers to this sample will help disentangle the roles of developmental plasticity, selection, genetic drift and gene flow on these morphological traits.

## ACKNOWLEDGEMENTS

Distribution data was supplied by Butterflies for the New Millennium which is run by Butterfly Conservation with funding from Natural England. We also thank Diana Taylor-Cox, Daniel Egleston, Sean Clough, Romain Villoutreix, Duncan Lister, Alex Elsy, Georgina Palmer and Melanie Smee for their help in sampling sites in the summers of 2016–2018, and Claire Williams for dissecting and photographing individuals collected in 2016/17. Thank you to Carl Yung for his advice and help with the field equipment. We thank Jon Bridle, Rus Hoelzel, Daniel Moore, Konrad Lohse, Stuart Piertney, Mario Vallejo-Marin, Aimee Walker and Christine Balmer for providing temporary space in their −80 °C freezers and liquid nitrogen to top up the dry shipper. Finally, we thank Steven Van Belleghem for his help with Patternize and adapting the package for our calibrations.

### Funding

This work was funded by the Natural Environment Research Council (NERC ACCE: studentship to Evelyn D. Taylor-Cox, grant number NE/L002450/1, NE/N015711/1 awarded to Ilik J. Saccheri and NE/N015797/1 Jane K. Hill). The funders had no role in study design, data collection and analysis, decision to publish, or preparation of the manuscript.

### Grant Disclosures

The following grant information was disclosed by the authors:
Natural Environment Research Council: NE/L002450/1, NE/N015711/1, NE/N015797/1.

### Competing Interests

The authors declare there are no competing interests.

### Author Contributions

- Evelyn D. Taylor-Cox and Ilik J. Saccheri conceived and designed the experiments, performed the experiments, analyzed the data, prepared figures and/or tables, authored or reviewed drafts of the paper, and approved the final draft.
- Callum J. Macgregor and Amy Corthine performed the experiments, authored or reviewed drafts of the paper, and approved the final draft.

- Jane K. Hill conceived and designed the experiments, authored or reviewed drafts of the paper, and approved the final draft.
- Jenny A. Hodgson conceived and designed the experiments, analyzed the data, authored or reviewed drafts of the paper, and approved the final draft.

## Field Study Permissions

The following information was supplied relating to field study approvals (i.e., approving body and any reference numbers):

Approvals were granted on a site-by-site basis via email or verbal communication from landowners. Permission was granted from: Durham Wildlife Trust, Forestry Commission (England and Scotland), Natural England, National Trust (England and Scotland), Norfolk Wildlife Trust, Scottish Natural Heritage, Scottish Wildlife Trust, Shropshire Wildlife Trust, Woodland Trust, Yorkshire Wildlife Trust, as well as, local/city councils, private land owners, private companies and site rangers.

## Data Availability

Images are available at Zenodo: Evelyn D. Taylor-Cox, Callum J. Macgregor, Amy Corthine, Jane K. Hill, Jenny A. Hodgson, & Ilik J. Saccheri. (2020). Wing morphological responses to latitude and colonisation in a range expanding butterfly. Zenodo. http://doi.org/10.5281/zenodo.3816842

Scripts and data are available at GitHub: https://github.com/EveTC/Pararge_aegeria_morphometrics.git

## Supplemental Information

Supplemental information for this article can be found online at http://dx.doi.org/10.7717/peerj.10352#supplemental-information.

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
