# Peer review of "Wing morphological responses to latitude and colonisation in a range expanding butterfly"

_PeerJ, doi:10.7717/peerj.10352_

## Round 0.1 · original submission · Major Revisions

The manuscript has now been thoroughly reviewed by two experts. Both of them acknowledge the impressive dataset.

Reviewer 2 is, overall, positive, but points out - besides many detailed, but important issues - the potential issue of multicollinearity of predictor variables.

Reviewer 1 is more critical, especially with regard (1) to the impossibility to differentiate between genetic and plastic effects, (2) to unjustified conclusions, and (3) to the novelty of the research. He also points out many additional, detailed issues in the carefully annotated manuscript.

While I am a bit less concerned about the novelty, I would like to see that the authors make an effort to refer to similar work (especially if it is on the same species!) and how their results are (not) in line with these previous results.

For all other concerns, the two reviewers have my full support. However, if the authors make an effort to carefully address all (!) points raised, I will be willing to look at the revised manuscript (probably after another round of reviews). I look forward to the resubmission!

Reviewer 1 ·

Basic reporting

The English is mostly clear and unambiguous. For suggestions for improvements please see annotated MS.
The background needs to be clarified, by paying more attention to plastic versus genetic variation. Hypotheses need to be clarified also. For some data, the relevance for the questions addressed is unclear.
Figures and Tables can be optimized, for details please see annotated MS.

Experimental design

Methods are sufficiently described. However, not all research questions are well defined. It was also not clear to me which gap of knowledge is targeted, as there are earlier studies on P. aegeria but also various other species on similar matters.

Validity of the findings

Replication is fine. I have several problems with the interpretation of statistical results and the conclusions drawn. Please see annotated MS for details.

Additional comments

The authors collected an impressive set of data, which has been analysed with sophisticated methods. Nevertheless, I have several concerns the authors may wish to consider. A major weakness of the paper is that it is impossible, using field-collected individuals, to distinguish among genetic and plastic effects. This is only briefly mentioned in the discussion. The study is also quite narrow in focus, and several previous study have addressed the same questions even using the same model. For details, please see annotated manuscript. My most important concerns are:

The Introduction needs to be rewritten. Please make a clear distinctions between plastic and genetic sources of variation. State which source of variation you are targeting, and which law / rule / prediction relates to plastic or genetic effects. Some concepts are not correctly or sufficiently explained in the introduction. It needs to be clear for the reader which type of variation you are targeting and what your predictions are. Currently, the hypotheses are partly very weak and do not follow from what is given in the introduction.

The authors should pay more attention to the implications of morphology for dispersal. Although often assumed, what is actually the evidence for a strong link between the two? Other factors (intrinsic ones, weather etc.) may be much more important. Supporting evidence should be given in the Introduction. Much more information is needed on the relevance of wing size and especially shape and colour.

I had repeatedly problems with the presentation of results, and especially the interpretation of statistical results. Several times, significant findings are claimed while this is not the case, or statistical trends are considered significant results. The whole result section needs a thorough revision. Please restrict yourself to statistically significant results. Based on these problems, I think that some interpretations and conclusions are not supported by your data.

The discussion should focus on the own results and put them into context. Currently, large parts give general information on the traits considered.

Annotated reviews are not available for download in order to protect the identity of reviewers who chose to remain anonymous.

Reviewer 2 ·

Basic reporting

The authors present a detailed and coherent study on the morphological responses of Pararge aegeria to range expansions, which is comprehensively written, well-structured and provides a sufficient background of the study question. Yet I, not having English as mother tongue, often struggled to get the meaning of the sentences. This is a pity, because it seems as if the authors present some remarkably interesting ideas, which are concealed by unprecise wordings. Mostly, my struggle results from an over-complication of sentence structure with the addition of highbrow words, which are not always necessary and/or could be replaced by simpler and more frequently used words. Thus, I think the whole text would benefit from being written in a simpler, more straight-forward manner, with shorter sentences and consistent use of terms. Please refer to the minor points in “general comments” for examples where I did not (directly) understood the authors statements.
In best scientific spirit, the authors not only provide their data, but also their scripts and macros. Unfortunately, the zenodo-link to the photographs which would have been needed to run the analysis did resulted in an internal server error.

Experimental design

The research question is well defined. However, the authors could point out the reasoning for investigation more clearly. The notion, that “climate-driven range expansions often covary with latitude (and temperature), making it hard to disentangle the effects of these processes” (i.e. demographic and environmental impacts on morphological traits), line 145-146, is very interesting and disentangling the effects seems to be the most important contribution of this study. However, this notion gets lost in the shuffle of the introduction. It deserves a more prominent position and surely, a greater resumption in the discussion.
Concerning the assessments of the used morphometrics and environmental variables, I applaud to the technical effort and detailed description thereof.
Four variables are used as independent variables: latitude, years colonized, mean developmental temperature and 10 year mean annual temperature. From the nature of those variables, I would expect issues with multicollinearity, and I don’t see whether such problems were looked at or addressed. Thus, the results might be highly biased. If, however, the authors considered multicollinearity and did check for it (which seems to be the case according to their R-script) , I strongly recommend adding a pairs plot of the chosen variables or something similar in the Appendix.

Validity of the findings

The authors discuss their results very carefully, and I really like this refreshingly non-over-selling. Maybe this had the unfortunate side effect, that the conclusions were not as concise as they could be; as stated in the “Experimental design section”, it would benefit from a more clear and basal discussion of the relative contribution of demographic and environmental effects on morphometrics.
There are some methodological / statistical issues which need to be addressed or discussed, specifically: multicollinearity, peaks close to the end or beginning of a month (see general comment Line 211-216), variation of developmental time along latitude (see general comments Line 216-220), and size standardization (see general comments Line 247-250).
Moreover, the authors should consider that they could not include the colour lightness of the body – probably the most important part in thermoregulation (which might explain the contrasting results to Zeuss et al. 2014)

Additional comments

Discussion in general: You make some good points. However, you often present them block wise, e.g. from line 451 to 467, you compare your found slope to that of similar studies. Then, from line 169-484 you outline the ecological effect of a certain shape in general and in line 186 – 496 you are more specific about the behaviour of P. aegeria. However, the latter two blocks are not directly connected to your results anymore. It would be better to link them more closely and be more straight forward. i.e. result: rounded forewing in younger populations -> the contrast of what one would suspect when looking at migratory species -> maybe because of the behaviour in P. aegeria

Considering that the divergence in cream-coloured spot and brown background colour of the wing takes considerable space, the reasoning behind this divergence is superficially introduced.

Line 60: Please replace “thermal tolerance” with “colour lightness”, given that Zeuss et al. (2014) did test the phylogenetic component of colour lightness but not thermal tolerance (in terms of temperatures which the species can cope with) per se .

Line 62-63: I got lost in this sentence. The relative importance of which response mechanisms? Please restructure the sentence to be less complicated for people like me who don’t have English as mother tongue.

Line 69: Maybe “is a product of” instead of “is predicted by”?

Line 74-75: I do not get this sentence.

Line 75-77: When you talk about dispersal traits before, and then mention Bergmann’s rule, you must point why a pattern in body size is connected to dispersal (i.e. body size is regarded as a dispersal trait)

Line 85-87: On first view, you give contradicting statements here. In parentheses, you give a clear direction (smaller individuals at higher temperatures) but then you say: “less clear is how the […] slope of this reaction norm is likely to be affected by temperature selection”. If “reaction norm” refers to the relationship between developmental temperature and body size, then you already have the slope, don’t you? Or is temperature selection something different than developmental temperatures? If the latter, can you be clearer what you mean with “temperature selection”?

Line 91-93: “thermal reaction norms may evolve in different directions” – can you call it a norm, if it is not consistent? I really struggle with this term.

Line 111-114: I would suggest moving this sentence behind the following, so that the sentences dealing with the TMH (general description and thermal properties of wings) are not interrupted by other mechanism descriptions.

Line 134: which range are you referring to? Dispersal?

Line 146-148: I did not get the point of this sentence

Line 211-216: This is an interesting and smart approach. However, I am wondering how much variation in Julien day of peak there is within 10 years. As you used month as a an unit to get the developmental temperatures, there might be problems if the sampling dates are close to the end or beginning of a month: For example, if you have the peak on the edge between June and July, it will make a big difference which month to choose, as the former then includes a quite cold month (April), the latter doesn’t. Maybe using the temperature of 90 Julian days would be a more suitable approach? It would be nice to see a graph in the Attachment depicting the phenology of the butterflies based on your data.

Line 216-220: Assuming a steady development time even under different developmental temperatures is seems problematic: as you mentioned yourself, southerly populations have more generations, and should have, at least under the temperature-size-rule, shorter development times. Hence, including also three months prior to emergence peak in southern generations will result in an underestimation of the developmental temperatures that were actually experienced.

Line 235-237: That’s a good point. Here, it would be good to say more about how well the recorder effort and the geographic coverage of the records was. For the latter, it would be of utmost importance to have a good, consistent geographic coverage when dealing with range-expansion. For example, the gap between the English and Scottish populations could be a sampling effect (i.e. not enough people which participated in the surveys living there) – unlikely, but it would be better to really see some sampling-coverage-map in the Appendix to be assured that it is no sampling effect.

Line 247/ Line 250: If the method used standardises specimens through controlling size, how can centroid size be used as a measure of wing size?

Line 309: Why did you exclude years since colonization? According to Mattila (2015) also dispersal might be influenced by melanism. Thus, I do not understand why you handle body size and melanism differently.

Line 345: The question is, whether 1.5% and 2.5% of change in wing size are biologically meaningful

Line 352-353: What do you mean with “Individuals […] fall along the same regression line”?

Line 392-394: if the cream has a steeper increase in colour lightness compared to the brown, than the slopes can’t be parallel

Line 397-398: Put the reference to Dennis and Shreeve (1989) to the discussion

Line 436: what do you mean with “they”?

---

## Round 0.2 · Minor Revisions

Thank you very much for your thorough revision of the manuscript! One of the reviewers of the first round has had a close look again and was generally happy with the result of the revision, and so was I.
However, the reviewer has raised a few additional points which you may want to address in a last round of minor revisions. I look forward to receive your final version of the manuscript!

Reviewer 2 ·

Basic reporting

no comment, everything is fine

Experimental design

great experimental design

Validity of the findings

everything fine

Additional comments

The authors have submitted a thoroughly revised manuscript and apparently, have put large effort into addressing the raised concerns. It reads now very well, especially the introduction. I now could follow every sentence even before my first morning coffee! (Excuse my non-professionality here, but that really means they did a good writing). In circumstances were issues or suggestion raised by the reviewers were not implemented, the authors expressed fair reasons why this was not possible or desirable from their point of view. Overall, this manuscript was good to start with based on the detailed methods, but now even got better due to a profound writing style and argumentation. There are only minor things with which I struggle, but I am confident that the authors will continue their superb efforts in addressing them without any problems.
1) The cream spot hypothesis.
There is still a part in the introduction missing, why the authors (or Dennis & Shreeve, 1989) are expecting cream spots to become larger with latitude.

Dennis & Shreeve have some interesting ideas about this hypothesis, which might also add substance to the reasoning for the divergence in cream-coloured spot and brown background colour. I have extracted some passages which I think could be especially useful:
- “expectations are that as summer conditions become cooler and less predictable to the north and west, selection has resulted in modifications in adult phenotype in response to thermoregulatory requirements and the needs of mate advertisement and predator escape.”
- “Cooler, cloudier conditions result in the lower activity of adults (rate, duration), which in turn influences their ability to locate mates, to court conspecifics successfully, to release their egg load, to nectar efficiently, to escape predators and to colonize new habitats. Depression of any of these biological functions adversely affects reproductive success. Some modifications in phenotype are known to be advantageous in cool climates. These changes depend on species-specific basking modes.”
- “The upper wing surfaces in dorsal baskers are often darker at the base and outer margin, but have more extensive brighter […] areas in the outer portion of the wings, which also allows the larger apical ocelli to be highlighted […]”

2) The discussion might benefit from a paragraph or sentences which discuss which effect demographic impacts on wing morphology might have under the expected change in environmental conditions. Let’s take the striking results for body size (being both shaped by range expansion process and latitude, but not subjected to temperature-size rule) as an example. While the authors carefully evaluate why there might been a mismatch between the latter two effects (latitude vs developmental temperature), a discussion of the range expansion effect in combination with latitude and developmental temperature could raise also interesting questions. E.g., does the Scottish South-East expansion (and thus selection for larger-bodied animals in south east) outdo the latitudinal effect? One or two sentences per trait would do to underline the important statement drawn in the conclusion: “Interpretation of the patterns observed here [and elsewhere!] must take account of the dynamic nature of this recent and ongoing range expansion”.

---

## Round 0.3 · accepted · Accept

Thank you very much for revising the manuscript according to the reviewer's comments! I look forward to seeing the paper published!